# Long-Term Changes in Tibial Plateau Angle (TPA) Following Tibial Plateau Leveling Osteotomy (TPLO) in Dogs—A Retrospective Study

**DOI:** 10.3390/ani14223253

**Published:** 2024-11-13

**Authors:** Magdalena Morawska-Kozłowska, Yauheni Zhalniarovich

**Affiliations:** Faculty of Veterinary Medicine, Department of Surgery and Radiology with the Clinic, University of Warmia and Mazury in Olsztyn, 10-718 Olsztyn, Poland; eugeniusz.zolnierowicz@uwm.edu.pl

**Keywords:** tibial plateau angle, cranial cruciate ligament, canine surgery, long-term outcomes, stifle stability

## Abstract

This retrospective study examines how the tibial plateau angle (TPA) changes over time in dogs following a tibial plateau leveling osteotomy (TPLO), a common surgery for treating cranial cruciate ligament (CCL) ruptures. The TPLO procedure stabilizes the stifle joint by rotating the tibial plateau to reduce stress on the ligament and improve joint function. While short-term outcomes of this surgery are well-documented, this study focuses on how the TPA evolves over a longer period, up to 12 months after surgery. We studied 60 dogs of various ages and weights, measuring the TPA before surgery, immediately after, and at 8 weeks, 6 months, and 12 months post-surgery. Our findings showed a steady increase in TPA over time, indicating that the tibia continues to remodel long after the initial healing period. No dog showed a decrease in the TPA, with the average TPA increasing from 4.98 degrees post-surgery to 9.02 degrees at 12 months. These results highlight the importance of long-term monitoring and suggest that aiming for a lower TPA immediately after surgery may result in better long-term outcomes for dogs undergoing TPLO.

## 1. Introduction

Surgical treatment for a ruptured cranial cruciate ligament (CCL) focuses on preventing the development of degenerative joint disease and aims to restore the stifle joint’s normal biomechanics [1]. There are various surgical techniques for addressing a ruptured CCL, and choosing the most effective method remains debatable [2]. Techniques that involve osteotomy and altering the tibia’s geometry have shown high success rates in stabilizing the stifle joint during movement [3]. In 1984, Slocum and Devine introduced a surgical approach called cranial tibial wedge osteotomy (CTWO), which involved decreasing the tibial plateau angle (TPA) [4]. However, due to a high rate of complications, CTWO was later replaced by tibial plateau leveling osteotomy (TPLO), which avoids altering the tibia’s length and does not require a full bone shaft osteotomy [2]. The TPLO method, described by Slocum in 1993 [5], involves performing a ¼-circle osteotomy of the proximal tibia and rotating the osteotomized fragment to adjust the tibial plateau angle. The desired postoperative TPA can be achieved by calculating the necessary rotation angle from the preoperative TPA, typically around 6.5 ± 0.9 degrees [6]. Following the leveling of the tibial plateau, a bone plate and screws are employed to stabilize the osteotomy during the bone healing process [7].

The current literature predominantly examines the TPLO technique, its clinical outcomes, and potential complications following the surgery [8]. However, there is limited focus on the long-term effects of the treatment [9,10], and no existing reports explore changes in the TPA during long-term (twelve months) follow-up recovery. De Souza et al. examined tibial plateau angle (TPA) measurements in 32 dogs at three critical stages: pre-surgery, immediately post-surgery, and after the tibial healing. The precise timing of the third measurement—taken during the tibial healing—was not explicitly defined [10]. Moeller et al. also investigated changes in the tibial plateau angle (TPA) during post-operative recovery following tibial plateau leveling osteotomy (TPLO). In this study, however, the timing of TPA measurements varied among patients, with assessments conducted between 28 and 65 days after the procedure [9]. This study aims to assess whether and how the TPA in dogs changes during the healing process following TPLO. Assessing the change in the TPA is crucial for evaluating whether the widely accepted standards for the TPA should be revised following the TPLO procedure [11].

## 2. Materials and Methods

This study had a retrospective design. The study group comprised sixty patients from the Department of Surgery and Radiology at the Clinic of the Faculty of Veterinary Medicine, University of Warmia and Mazury in Olsztyn. Each animal was diagnosed with a complete unilateral rupture of the cranial cruciate ligament (CCL). The diagnosis was confirmed using positive results from the drawer and tibial compression tests and mediolateral radiographs of the affected limbs. Dogs with bilateral conditions and those with partial cranial cruciate ligament ruptures were excluded from the study. The TPLO surgical technique was chosen as the treatment method in all patients. The unilateral TPLO procedure was conducted on 60 mature dogs, ranging in body weight from 7 kg to 59 kg and between 2 and 8 years of age, by one veterinary surgery specialist.

Data were gathered from 60 dogs that experienced cranial cruciate ligament tears and had single-side TPLO surgery. Following the initial diagnosis, the next step was to radiologically evaluate the mediolateral radiographs of the stifle joints, including the hock joints, and determine each patient’s preoperative tibial plateau angle (TPA). The X-rays were taken and evaluated by the same person. For this purpose, three lines were marked on each radiograph:The line of the tibial plateau slope runs through two landmarks: the cranial attachment of the CrCL (cranial cruciate ligament) and the distal attachment of the CaCL (caudal cruciate ligament).The line of the mechanical axis of the tibia runs between the center of the intercondylar eminence of the tibia and the center of the tarsocrural joint.The line perpendicular to the mechanical axis of the tibia at the intersection of the other two lines (Figure 1).

The TPLO procedure was performed in all dogs according to contemporary guidelines using LCP systems [5,8]. The surgical approach began with a medial parapatellar skin incision, followed by blunt dissection of the subcutaneous tissues. In each case, a medial mini-parapatellar arthrotomy was performed, and meniscal tears were addressed through partial meniscectomy, hemimeniscectomy, or total meniscectomy, depending on the specific case. Access to the proximal tibia was gained through a medial approach, and the TPLO procedure was completed using locking implants as described by Slocum [12]. X-rays and TPA measurements were conducted. Postoperative radiographs were taken immediately after the procedure and at follow-up visits at eight weeks, six months, and twelve months post-TPLO.

The author performed all measurements using the vPOP-pro program (VetSOS Education Ltd. of Column House, Llangollen, UK). The obtained data were then classified, and the degree of change in the TPA over time was averaged.

Data requiring statistical analysis were collected in MS Word text files and MS Excel spreadsheets. Subsequently, statistical analysis was performed using the analysis of variance method with Dell Statistica software [TIBCO Software Inc., Polska, Poznań (2017); Statistica (data analysis software system), version 13]. The significance of differences between the mean TPA values in the animals was determined using the Tukey test (HSD) at the *p* = 0.05 level.

## 3. Results

Demographic information was collected from a sample of 60 dogs with body weights ranging from 7 kg to 59 kg and ages between 2 and 8 years. As part of this research, mediolateral radiographs of the stifle joints were taken at follow-up visits in the eighth week, the sixth month, and the twelfth month post-procedure. These radiographs were used to measure the TPA using the vPOP-pro program, and the change between the initial (right after the procedure) and final TPA (after twelve months) angles was calculated for each patient. The results are presented in the table below (Table 1).

Each patient exhibited an increased TPA within one year of the TPLO procedure, with no patient showing a decrease in the TPA. An increase of 4.7 degrees was observed in 21.7% of the patients (Figure 2, Table 3).

## 4. Discussion

Previous studies have determined the minimum TPA required after TPLO to alter the stifle joint geometry so that the cranial cruciate ligament bears a more significant portion of the forces acting on the joint [6,13,14]. Consequently, the theoretical success of TPLO relies on achieving this minimum TPA to prevent cranial translation of the tibia relative to the femoral condyles. Ensuring tibial plateau rotation and maintaining this position during the bone healing process is essential for a successful clinical outcome and to preserve the proper biomechanics of the stifle joint [7].

When determining the TPA after TPLO at various time intervals, it is essential to consider that the tibia undergoes healing and remodeling postoperatively over time [15]. In the post-TPLO follow-up radiographic evaluations, specific features are assessed, such as the degree of rounding in the distal cortex at the osteotomy site, continuity of the two cortices, subjective assessment of periosteal proliferation and callus remodeling, as well as the visibility of the osteotomy line. This visibility may reflect not just the bone healing process, but the ongoing remodeling of the tibia over time [16]. All of these bone union factors can influence the TPA changes. In this study, the authors highlight that the tibial healing process plays a crucial role in altering the tibial plateau angle (TPA) during the long-term recovery of dogs following TPLO surgery. The consistent remodeling of the tibia, observed in all cases, was evident in the radiographic images, particularly through the rounding of the distal fragments along the osteotomy line (Figure 3B). This suggests that bone healing directly impacts the TPA, potentially influencing postoperative outcomes. However, the authors acknowledge the need for further investigation to account for other factors that could affect the extent of callus remodeling. These factors include the patient’s weight, diet, and rehabilitation regimen post-surgery, which were not controlled in this study (none of the patients underwent intentional physical or physiotherapy procedures; only movement restrictions were applied). Future research should aim to isolate these variables to provide a more comprehensive understanding of their influence on the healing process and TPA changes. This will help refine postoperative management strategies and potentially improve long-term outcomes in TPLO-treated dogs.

Studies show that factors like gender, age, and weight have a significant impact on the healing outcomes of tibial plateau leveling osteotomy (TPLO) in dogs. For example, a dog’s age and weight can influence postoperative recovery and the likelihood of complications, with older or heavier dogs tending to have longer recovery times and an increased risk of issues [17,18].

A limited number of studies report long-term follow-up concerning the TPA following TPLO surgery. In 2006, Moeller et al. documented changes in the TPA during tibial healing by analyzing data taken between 28 and 65 days post-op. The mean increase in TPA was 1.5 ± 2.21 degrees, ranging from −3 to 9 degrees. However, the authors acknowledged that their study did not include radiographic assessment after complete healing and bone formation, and the follow-up period was relatively short. The changes in TPA were attributed to incomplete patient immobilization postoperatively, and the authors concluded that the reasons for changes in TPA were mechanical rather than a result of biological tibial bone reconstruction. They also studied the factors that might contribute to the suspected movement of the rotated part of the tibia during the healing process, considering the patient’s weight and the type of TPLO plate used (locking/non-locking). Still, no statistically significant factors influencing the TPA were identified [7]. In 2009, a study was conducted to examine the relationship between TPA increases and the use of locking or conventional screws to fix the TPLO plate. Radiographs were taken immediately after surgery and again at eight weeks post-TPLO. The study found a more significant increase in TPA when non-locking screws were used [19]. In a more recent study, De Souza et al. (2021) examined changes in the TPA following TPLO procedure at three critical points: before surgery (TPA1), immediately following the TPLO procedure (TPA2), and once the bone had fully healed (TPA3). Notably, the timing of the final measurement varied among patients and was not standardized, which may have significantly influenced the study’s outcomes. The findings indicated only minor changes in TPA across the healing period, which were statistically insignificant, suggesting that adjustments in TPA during recovery might not directly affect TPLO success. However, the study’s small sample size and relatively short follow-up period (an average of 128 days) limited the strength of these conclusions. The authors suggested that while minor TPA changes occurred during healing, they did not appear to compromise the surgery’s stability or success, countering the idea that the “rock back” phenomenon has a detrimental effect [10]. In contrast, our study applies a consistent timeline for TPA measurements across all patients, with the final measurement taken one year post-procedure. Long-term radiological observations demonstrated that TPA continues to change after clinical bone growth has occurred and may cause a “rock back” effect. However, the limited number of follow-up measurements and the short study duration made it challenging to confirm the hypothesis conclusively. In our study, patients underwent TPLO using locking systems, which may have influenced the results.

In another study, the impacts of limb positioning and X-ray central beam orientation on TPA measurements were evaluated [7,20,21]. It was found that both factors significantly influenced the results. The authors suggest that the central X-ray beam should be accurately aimed at the cranial edge of the stifle joint. Any misalignment of the central beam can distort the radiographic view of the stifle, which in turn affects the precision of TPA measurements. If the beam is positioned too proximally toward the tibial fragment, the TPA appears larger, while caudal or distal beam placements result in a lower angle. Furthermore, improper limb positioning, such as misalignment of the femoral condyles, can cause TPA measurement discrepancies of up to 3.61 degrees compared to radiographs taken with proper joint alignment [20]. In our study, pre-and post-operative radiographs were performed under sedation or general anesthesia, ensuring proper limb positioning and accurate central beam orientation directly over the cranial border of the stifle joint. All patients achieved correct flexion of the stifle and ankle joints and proper overlap of the femoral condyles. Moreover, the radiographs were consistently taken by the same individual, which contributed to the precision of the TPA measurements.

A key factor influencing variations in TPA measurements is the subjective interpretation of radiographs by different individuals, resulting in differences of up to 1.51 degrees [21,22]. In this study, however, the same person performed all TPA measurements using the same technique and materials for each examination. This approach dramatically reduces the potential for error and improves the consistency of measurements throughout the procedure.

This study has several limitations because of its retrospective nature. Due to the relatively small research group- breed and sex were not analyzed in this study. Another limitation is that changes in TPA over time may be influenced by the progression of osteoarthritis, which can affect the appearance of anatomical landmarks used for measurement.

## 5. Conclusions

The results indicate a statistically significant change in the TPA not only during the healing of the osteotomy line but also after the tibial reconstruction process is completed. The observed increase in TPA values across patients suggests that biological bone tissue remodeling substantially alters tibial geometry. Furthermore, a correlation was noted between the TPA immediately after the procedure and the TPA one year later for each patient. Specifically, a smaller TPA immediately post-procedure was associated with a smaller difference between the final (after twelve months) and initial TPA (right after the procedure). If this relationship is validated in a larger patient cohort, it could suggest establishing new guidelines for the rotational angles of the proximal tibia during TPLO procedures. While many studies focus on long-term clinical outcomes, this study is notable for being the first to present long-term radiographic outcomes after TPLO. In contrast, previous reports have typically focused on shorter recovery periods, usually up to eight weeks.

## Figures and Tables

**Figure 1 animals-14-03253-f001:**
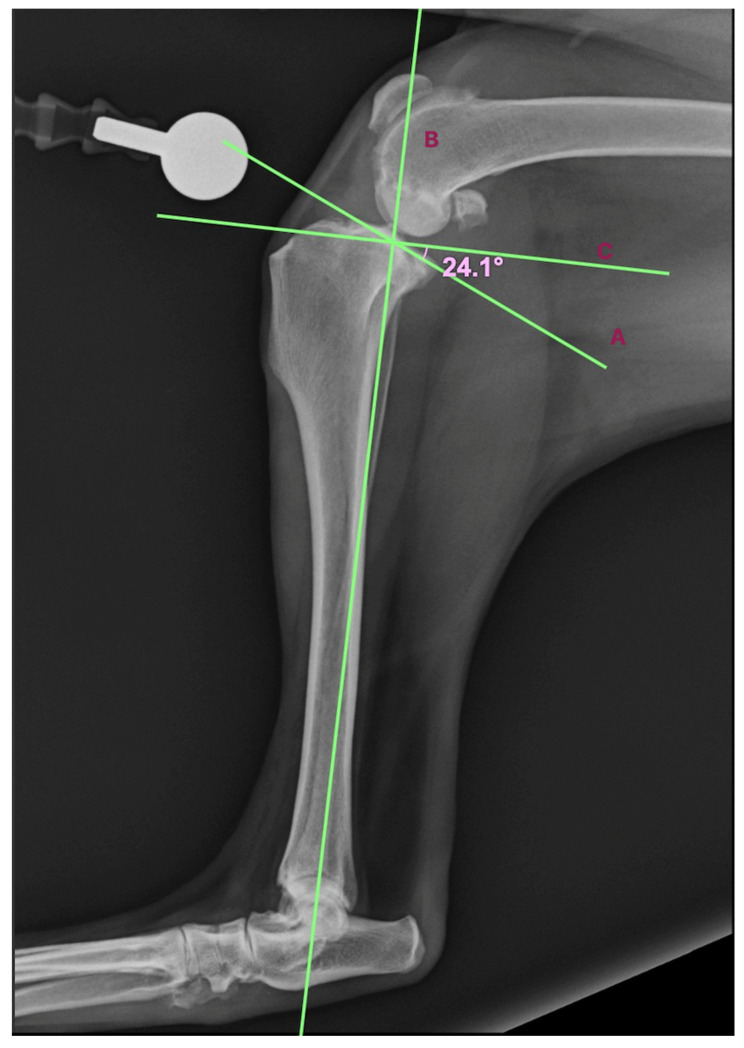
The mediolateral radiograph shows preoperative TPA measurements, where TPA is the acute angle between lines A and C.

**Figure 2 animals-14-03253-f002:**
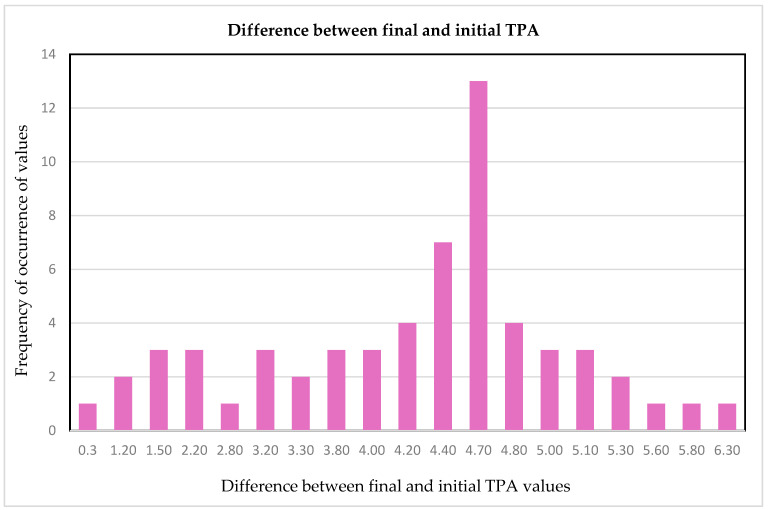
This graph illustrates the relationship between the initial and final TPAs and the frequency of each value among all patients who underwent the TPLO procedure.

**Figure 3 animals-14-03253-f003:**
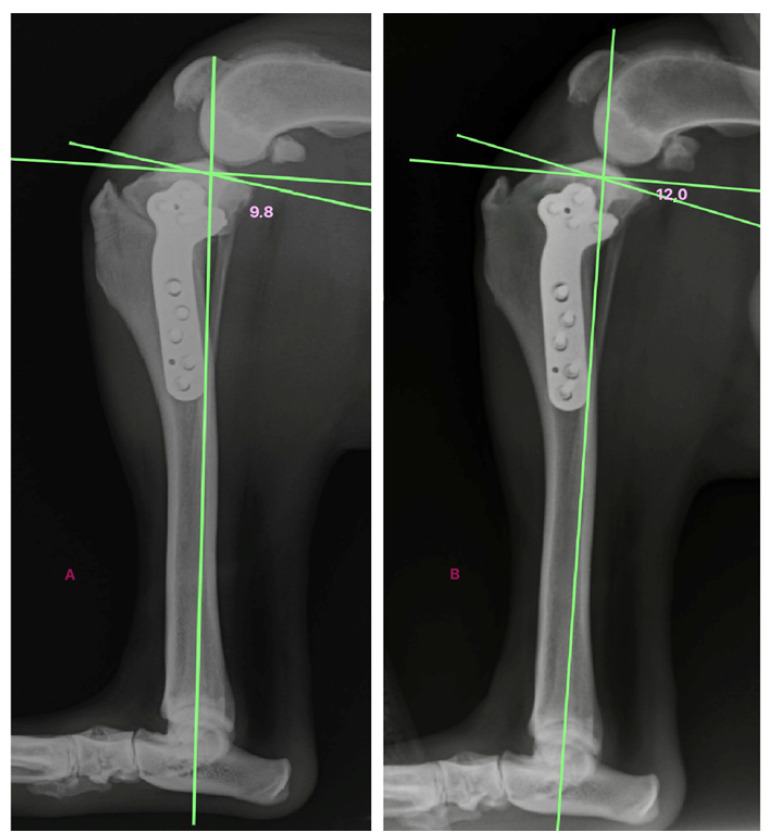
TPA measurements in Dog 6. (**A**) TPA measured directly after surgery. (**B**) TPA measured one year post-surgery.

**Table 1 animals-14-03253-t001:** TPA measurements after TPLO treatment.

	TPA (°)				
	Immediately Post-Op	Eight Weeks After Surgery	Six Moths After Surgery	One Year After Surgery	Difference Between Final and Initial TPA
Dog 1	5.4	6.9	8.9	10.1	4.7
Dog 2	5	5.9	9.2	11.3	6.3
Dog 3	7.1	8.9	10.3	10.3	3.2
Dog 4	3.7	5	6.4	8.1	4.4
Dog 5	5.4	6.7	8.6	9.8	4.4
Dog 6	9.8	10.9	11.8	12	2.2
Dog 7	9.2	10.4	11.3	11.4	2.2
Dog 8	4.5	5.5	6.4	7.8	3.3
Dog 9	1	2.4	2.4	2.5	1.5
Dog 10	5.2	5.9	6.2	6.7	1.5
Dog 11	3	3.8	4.1	4.2	1.2
Dog 12	6.1	7.6	9.3	9.9	3.8
Dog 13	3.9	4.9	5.8	8.1	4.2
Dog 14	2.1	3.5	3.8	4.3	2.2
Dog 15	5.1	6.3	8.1	9.8	4.7
Dog 16	6.4	7.1	9.1	10.4	4
Dog 17	3.9	5.4	6.9	7.2	3.3
Dog 18	4.9	5.2	5.2	5.2	0.3
Dog 19	5.3	6.8	8.4	10	4.7
Dog 20	4.7	6.8	7.6	9.7	5
Dog 21	5.1	7.2	8.1	10.2	5.1
Dog 22	6.1	8.3	9.12	11.2	5.1
Dog 23	9.1	11.2	11.5	13.3	4.2
Dog 24	3.4	4.5	6.7	8.1	4.7
Dog 25	4.8	6.2	7.5	9.8	5
Dog 26	5.5	7.1	8.2	10.3	4.8
Dog 27	5.9	7.4	8.9	11	5.1
Dog 28	3.1	3.5	4.2	4.3	1.2
Dog 29	4.3	6.1	8.3	10.1	5.8
Dog 30	5.1	6.2	8.7	9.8	4.7
Dog 31	4.6	6.8	9.1	10.2	5.6
Dog 32	3.9	4.5	5.2	8.9	5
Dog 33	4.3	5.2	7.1	9.1	4.8
Dog 34	6.9	7.3	8.9	11.3	4.4
Dog 35	3.5	4.1	5.6	8.2	4.7
Dog 36	5.2	6.2	7.5	9.9	4.7
Dog 37	5.3	5.9	6.5	9.1	3.8
Dog 38	6.1	6.3	7.5	10.1	4
Dog 39	4.8	5.3	6.2	9.2	4.4
Dog 40	3.1	3.9	4.8	6.3	3.2
Dog 41	2.9	3.1	4.2	4.4	1.5
Dog 42	7.2	9.2	11.3	12.5	5.3
Dog 43	4.5	5.4	6.9	9.2	4.7
Dog 44	4.7	5.9	7.1	9.4	4.7
Dog 45	5.8	7.4	9.8	11.1	5.3
Dog 46	3.9	4.2	6.5	8.3	4.4
Dog 47	4.1	4.9	5.6	7.3	3.2
Dog 48	4.6	6.7	7.9	9.3	4.7
Dog 49	6.2	7.3	8.4	10.9	4.7
Dog 50	4.6	6.2	7.3	9.4	4.8
Dog 51	3.9	4.8	5.6	6.7	2.8
Dog 52	5.4	6.9	7.8	9.6	4.2
Dog 53	5.7	7	8.5	10.1	4.4
Dog 54	5.1	6.3	7.4	9.5	4.4
Dog 55	4.3	5.2	7.6	9	4.7
Dog 56	6.1	7.3	9.5	10.9	4.8
Dog 57	3.8	4.6	5.9	8	4.2
Dog 58	5.3	6.9	7.8	10	4.7
Dog 59	4.8	5.7	7.1	8.6	3.8
Dog 60	4.2	5.3	6.5	8.2	4
Average	4.98	6.15	7.47	9.02	4.04

The data were analyzed statistically using Statistica (data analysis software system), version 13. The results indicated that the mean postoperative TPA was 4.98 degrees. At follow-up visits at eight weeks, six months, and twelve months after the procedure, the mean TPAs were 6.15, 7.47, and 9.02 degrees, respectively. The average difference between the final and initial TPAs was 4.045 degrees (median 4.4 degrees; range 0.3 to 6.3 degrees) (Table 2). This difference was statistically significant (*p* = 0.05).

**Table 2 animals-14-03253-t002:** Statistical analysis of the obtained test results is used to determine the change in the TPA over time after the TPLO procedure.

Statistics		
Difference Between Final and Initial TPA		
N	Important data	60
	Important data	0
Average		4.0450
Median		4.4000
Data dominant		4.70
Standard deviation		1.25879
Minimum		0.30
Maximum		6.30
*p*-value		0.05
TPA after surgery		
Average		4.98
Standard deviation		1.53970
*p*-value		0.05
TPA eight weeks after surgery		
Average		6.15
Standard deviation		1.748352
*p*-value		0.05
TPA six months after surgery		
Average		7.47
Standard deviation		1.955343
*p*-value		0.05
TPA one year after surgery		
Average		9.02
Standard deviation		2.152009
*p*-value		0.05

**Table 3 animals-14-03253-t003:** This table presents the statistical analysis of the differences between the final and initial TPAs obtained after the TPLO procedure.

Difference Between Final and Initial TPA Values					
		Frequency of Occurrence	Percent	Percentage of Valid	Cumulative Percentage
Data	0.3	1	1.7	1.7	1.7
	1.20	2	3.3	3.3	5.0
	1.50	3	5.0	5.0	10.0
	2.20	3	5.0	5.0	15.0
	2.80	1	1.7	1.7	16.7
	3.20	3	5.0	5.0	21.7
	3.30	2	3.3	3.3	25.0
	3.80	3	5.0	5.0	30.0
	4.00	3	5.0	5.0	35.0
	4.20	4	6.7	6.7	41.7
	4.40	7	11.7	11.7	53.3
	4.70	13	21.7	21.7	75.0
	4.80	4	6.7	6.7	81.7
	5.00	3	5.0	5.0	86.7
	5.10	3	5.0	5.0	91.7
	5.30	2	3.3	3.3	95.0
	5.60	1	1.7	1.7	96.7
	5.80	1	1.7	1.7	98.3
	6.30	1	1.7	1.7	100.0
	Total	60	100.0	100.0	

## Data Availability

Data will be provided upon request.

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
