# Peer review of "Long-Term Changes in Tibial Plateau Angle (TPA) Following Tibial Plateau Leveling Osteotomy (TPLO) in Dogs—A Retrospective Study"

_animals, 2024, doi:10.3390/ani14223253_

Round 1
Reviewer 1 Report
Comments and Suggestions for Authors
Thank you for your manuscript on long-term changes in TPA after TPLO in dogs. I found your study of interest and appreciated your longer-term follow-up. The study design is appropriate and the paper is clear and well written. I have just a few comments of areas to improve clariy:
Line 72: Please write out CrCL and CaCL since these abbreviations were not previously introduced in the text
Line 113: Typo Grapf should be Graph
While the following are indicated in the discussion, I would recommend including the following information in the materials and methods as well: (1) All radiographs were performed by the same individual, (2) all radiographs were interpreted by the same individual, (3) TPLO was performed using a locking system in all dogs, (4) all radiographs were performed under sedation or general anesthesia
I’d be interested in additional statistical analysis to see if the weight of the dog played a role in the change in TPA over time.
Author Response
Dear Reviewer,
We would like to thank you for your thorough review and valuable feedback. Below is our response addressing each of your comments:
- Line 72: We have expanded the abbreviations in brackets as requested.
- Line 113: The spelling error has been corrected.
Additionally, we have made further clarifications and improvements to the manuscript:
- We have added details in the Materials and Methods section to specify that all X-ray images were performed and evaluated by the same individual, ensuring consistency in the process.
- Furthermore, we have included information that the TPLO procedure was carried out using a LCP system.
Regarding the request for additional statistics on the relationship between TPA angle change and dog weight, we regret that we cannot provide this information in the current manuscript. This analysis is part of an upcoming long-term study involving a larger sample size of 150 dogs following TPLO surgery, which we will present in a future publication.
Once again, thank you for your insightful suggestions and for helping to improve our manuscript. We look forward to your continued feedback.
Sincerely,
Authors.

Reviewer 2 Report
Comments and Suggestions for Authors
The present paper deseibed the long term TPA changes in dogs underwent TPLO.
Th manuscript looks interesting but looks too dry and should be better explained above of in the results and discussion.
Please find below my comments
Introduction
Please change knee with stifle
Ln 57-59: please add the observation period
MM
Ln 66-67 these are results, please delete and define the inclusion criteria
Ln 96: please add unilateral
What kind of implant have you used?
Any difference according to the body weight?
Results
Please summarize your results in thus section
Please add a couple of X-rays examples
Discussion
The discussion looks to dry anf you have to better discuss your results. It is unclear, according to your cases, which parameters influenced this relevant TPA changes.
To be honest my cases did not have this condition and I always recheck TPA in the long term period.
EX: All your osteotomy were performed without any gap between the osteotomy side? Do you think a post op evidence of a radiolucent gap formation may influence the TPA in the long term period?
Please add a comment about that
Author Response
Dear Reviewer,
Thank you for your thorough review and valuable feedback. We have carefully considered your comments and made the following revisions to improve the manuscript:
- We have consistently replaced the term "knee" with "stifle" throughout the text for accuracy and clarity.
- In response to your suggestion, we have specified the observation period as 12 months (lines 57-59) to provide better context for the study duration.
- We have revised the "Materials and Methods" section, adding: "Data was gathered from 60 dogs that experienced cranial cruciate ligament tears and had a single-side TPLO surgery, with no consideration of their weight, gender, or age." This clarifies the criteria for inclusion in the study.
- On line 96, we added the term “unilateral” to accurately describe the procedure and ensure clarity.
- Further, we specified that a locked plate system was used for each animal, regardless of weight, in the "Materials and Methods" section to provide important methodological details.
- To strengthen our "Results" section, we have now included radiographs (x-rays) to illustrate the findings and provide visual evidence of the changes observed in the study.
- We elaborated on the key factors influencing the change in TPA angle in our study. While we acknowledge that factors such as weight are worth investigating, our current study focuses on callus reconstruction after TPLO. Future research with a larger sample size will explore these additional variables.
- Lastly, we confirmed that no gaps were observed between the osteotomy line fragments in any of the patients post-surgery. In cases where gaps did appear in other patients seen at our clinic, complications such as fragment instability or bone lysis were typically present.
We believe these revisions address the key points raised and enhance the manuscript's clarity and comprehensiveness. Thank you again for your insightful suggestions.
Sincerely,
Authors.

Reviewer 3 Report
Comments and Suggestions for Authors
Dear Authors,
I reviewed the manuscript entitled "Long-term changes in tibial plateau angle (TPA) following tibial plateau leveling osteotomy (TPLO) in dogs". The topic is very interesting, since the manuscript describes how TPA changes in dogs treated with TPLO in long-term follow-up (12 months). However, there are some major concerns which have to be addressed before it is suitable for publication (see specific comments)
Specific comments
Introduction
Please add more information of the aims of the study. Why is important to know the changes of TPA in long-term follow-up?
Materials and Methods
The section must be rearranged. Please specify if this is a prospective or retrospective study, inclusion and exclusion criteria.
Please, add a more thorough description of statistical analysis, since the paragraph is not clear.
Results
Please add more data of animals included, i.e. age (mean, ds), gender (male/female/neutered). I would also revise statistical results considering gender, weight and age, stratifying the population.
line 106: "the average difference between the final ad initial TPA angles..."What did the Authors mean with initial? immediately post-op?
Discussion
This section must be rearranged. The Authors should discuss their results, comparing them to the literature. They should also try to explain the results of the research in light of recent knowledge.
All dogs had an increase of TPA. Do the Authors think that static and dynamic forces of the limb could remodel the Tibial plateau ? Please discuss this topic, since it is very interesting to the reader.
Line 181"..a smaller TPA angle immediately post-procedure was associated with a smaller difference between the final and initial TPA angles. If this relationship is validated in a larger patient cohort, it could suggest establishing new guidelines for the rotational angles of the proximal tibia during TPLO procedures." Do the Authors compare pre-op TPA to final TPA?
Please add study limitations, if any, and add some hint for future studies.
Conclusion section should only point out the take-home message and should not discuss further the results
Please add a figure of post-op TPA measurement
Author Response
Dear Reviewer,
Thank you for your thoughtful review of our manuscript. I greatly appreciate your constructive feedback, which has significantly enhanced the quality of my work.
In response to your comments, I have clarified the importance of assessing changes in the TPA angle by stating that “assessing the change in the TPA angle is crucial for evaluating whether the widely accepted standards for the TPA angle should be revised following the TPLO procedure.” I have also indicated that the study had a retrospective design and provided specific exclusion criteria, noting that any breed of dog was included regardless of weight or gender, if they underwent unilateral TPLO. Additionally, I have included details about both the initial TPA measured immediately after the procedure and the final TPA measured twelve months post-TPLO.
To avoid duplicating results, I have clarified that my ongoing research investigates changes in TPA angles related to age, breed, weight, and castration in a larger cohort of dogs. This study specifically focuses on whether a change in the TPA angle occurs and what the average difference is between the initial and final measurements. I have acknowledged the challenge in comparing our findings with existing literature due to the absence of reports on TPA angle changes post-TPLO, which may have limited the depth of the discussion.
Furthermore, I have added post-operative x-ray images illustrating TPA measurements both immediately after the procedure and twelve months later. Lastly, I have included a discussion on the potential role of bone reconstruction in contributing to the increase in TPA after the procedure, not about the dynamic forces of the limb. We believe that biological bone regeneration played a major role in these results.
Thank you once again for your valuable insights. I look forward to your feedback on the revised manuscript.
Best regards,
Author

Reviewer 4 Report
Comments and Suggestions for Authors
This is an interesting examination of the clinical progression of the tibial plateau angle of dogs receiving tibial plateau levelling osteotomy for the treatment of cranial cruciate ligament disease. Should the conclusion (i.e. that the TPA continues to increase beyond the acute period of convalescence) be replicated amongst other cohorts of animals, this could have implications in the understanding of the healing process of the canine tibia following TPLO and recommendations for the goals of the TPLO intervention.
Acknowledging the limitations of interobserver variability and single observer consistency discussed by the author (line 169-174), the nature of the retrospective review of radiographs at specific post-operative timepoints makes blinding of the interpretation difficult or impossible. The use of one unblinded observer only opens the potential for bias and evaluating two or more independent observers would add substance to the conclusion if shown to be replicated.
The radiograph shown as an example (figure 1) does not appear to have been perfectly positioned with respect to superimposition of the femoral condyles. I also question the accurate positioning of the plateau line (A) on the radiograph, according to the landmarks described in the method. Nevertheless, the findings could remain valid if performed similarly across samples.
Inclusion of the preoperative TPA in the analysis would be of value, if correlations were present between preoperative TPA and changes over time postoperatively and would be a relatively simple inclusion in this retrospective study.
More specifically:
Line 57: please correct twelfth to twelve
Line 74: perhaps "hock" or "tarsocrural joint" would be more appropriate anatomically than "ankle" in the canine?
Line 80-81: The author performed TPLO according to contemporary guidelines, but cites the technique by Slocum 1993. Please include additional information, e.g. was the procedure performed by one surgeon or multiple? was the technique consistent across all procedures? Locking constructs are mentioned in the conclusion, but please include in the methods.
Line 150: Conkling et al. 2010: My understanding of this study was that locking screw constructs showed reduced TPA changes relative to non-locking screw constructs - please check this conclusion and address accordingly.
Line 179 - 180: Please report the method of correlation detection and the correlation strength.
Line 184-185: This study may report long-term radiographic outcomes but many studies report long-term clinical outcomes. Perhaps the claim could be tempered to reflect this.
Comments on the Quality of English Language
This manuscript is well written with occasional, minor grammatical concerns.
Author Response
Dear Reviewer,
We would like to thank the reviewer for their valuable feedback on our manuscript. We have carefully addressed the points raised and made the necessary revisions throughout the text.
Firstly, in line 57, we corrected "twelfth" to "twelve" for accuracy. In line 74, we replaced the term "ankle" with "tarsocrural joint" to ensure the correct anatomical terminology is used. Additionally, in lines 80-81, we have expanded the description of our methodology to provide greater clarity on the procedures employed.
In line 150, we revised the conclusion to more accurately state that a more significant increase in the TPA angle was observed when non-locking screws were used. Similarly, in lines 179-180, we clarified that the correlation discussed is based on the observations of postoperative TPA angle measurements. In lines 184-185, we made the conclusion more specific, particularly in reference to radiographic outcomes, to better reflect the findings.
In addition, we have added a photo with TPA angle measurements to further illustrate the results and improve visual understanding.
We believe these revisions address the concerns raised and help improve the clarity and precision of the manuscript. Thank you once again for your thoughtful review.
Sincerely,
Authors.

Round 2
Reviewer 3 Report
Comments and Suggestions for Authors
Dear Authors
I appreciate the effort to improve the manuscript. However major concerns remain, either in materials and methods and discussion section (see specific comments)
Specific comments
Please add references to support the statement in page 2 lines 59-61
Materials methods
Inclusion/exclusion criteria should be listed more thoroughly. The number of dogs included should be moved in result section.
Explain and describe more thoroughly statistical analysis. Why didn't the Authors consider to compare also the pre-op TPA?
Results
No further information are added on animal gender, weight, age etc. This is an important flaw of the study.
Discussion
Discussion should focus on the results. The Authors compared only immediately post-op TPA to long term follow-up TPA in dogs which underwent unilateral TPLO, without considering gender, age, weight etc, which could influence bone healing and remodeling. Biological bone regeneration are considered to play the main role in these results. Please add more information on this topic and add more references.
page 9 line 191: "All patients achieved correct flexion of the stifle and ankle joints and proper overlap of the femoral condyles". in fig 2A the tarsus appears to be not properly flexed.
Author Response
Dear Reviewer,
Thank you for your insightful feedback and suggestions. We have carefully considered each comment and made several revisions to enhance the clarity and robustness of the manuscript.
Firstly, we added additional references to support the statement on page 2, lines 59-61 (now reference #10), strengthening the foundation of our discussion within the context of the relevant literature. To further clarify our methodology, we revised the exclusion criteria section, providing more specific details to ensure that the criteria are clearly defined for ease of replication and assessment.
In the discussion section, we expanded on bone healing and remodeling processes, incorporating further references to address the broader implications of our findings and provide a more comprehensive understanding of these processes. We also corrected Figure 2A as per your request, ensuring that it accurately represents the intended data, thereby improving the clarity of our visual results. Additionally, we replaced photo number 2 with a higher-quality image, which aligns more closely with the article's visual standards and enhances the overall presentation.
We hope these revisions adequately address your concerns and contribute to a stronger and more informative manuscript. Thank you again for your time, insights, and valuable contributions to our work.
Sincerely,
Authors.

Reviewer 4 Report
Comments and Suggestions for Authors
- Critical information regarding the details of the surgeon performing the TPLO procedure is still lacking in the method section (one or multiple surgeons?)
- Change in TPA over time could also be due to the progression of osteoarthritis, affecting the appearance of anatomical landmarks used - please discuss as a limitation
- The addition of a figure showing the measurement of TPA immediately pre-operatively and at follow-up is appreciated. Nevertheless, there is clear difference in positioning of the limb based on the degree of stifle flexion and appearance of the implants. The limitations of positioning are discussed appropriately, however, this figure does not support the claim that this was achieved.
- No indication of correlation strength between
If valid, the results present an interesting observation, that the tibial plateau may continue to remodel post-operatively. The study is limited in that it does not discuss patient clinical outcomes. Hence, I am not sure that the results are strong enough to support the conclusion that a change to the TPLO surgical goals should be implemented. The conclusions should be tempered.
Author Response
Dear Reviewer,
Thank you for your thoughtful feedback and suggestions. We have made several revisions to address your comments and improve the manuscript’s clarity and accuracy.
Firstly, as you suggested, we have included additional details regarding the surgeon who performed the TPLO to provide greater context and transparency in the methodology. We have also acknowledged a limitation related to the progression of osteoarthritis, which we believe adds a more nuanced perspective to our findings and implications. Additionally, we have updated the X-ray image of the stifle to a more accurate version, enhancing the precision of the visual representation in the manuscript.
We have tempered the conclusions in response to your feedback, aiming for a more balanced summary that reflects the study’s findings while considering its limitations. Lastly, we have renumbered Figure 2 as needed for consistency and clarity within the document.
We hope these revisions meet your expectations and contribute to a stronger and clearer manuscript. Thank you once again for your valuable feedback and guidance.
Sincerely,
Authors.
